# Gliotic Response and Reprogramming Potential of Human Müller Cell Line MIO-M1 Exposed to High Glucose and Glucose Fluctuations [note 1]

**DOI:** 10.3390/ijms252312877

**Published:** 2024-11-29

**Authors:** Benedetta Russo, Giorgia D’Addato, Giulia Salvatore, Marika Menduni, Simona Frontoni, Luigi Carbone, Antonella Camaioni, Francesca Gioia Klinger, Massimo De Felici, Fabiana Picconi, Gina La Sala

**Affiliations:** 1Unit of Endocrinology and Diabetology, Isola Tiberina-Gemelli Isola Hospital, 00186 Rome, Italy; benedetta_russo6@msn.com (B.R.); marika.menduni@gmail.com (M.M.); 2Section of Histology and Embryology, Saint Camillus International University of Health Sciences, 00131 Rome, Italy; giorgia.daddato@alumni.uniroma2.eu (G.D.); klinger@uniroma2.it (F.G.K.); 3Department of Biomedicine and Prevention, University of Rome Tor Vergata, 00133 Rome, Italy; giulias@mclink.it (G.S.); camaioni@uniroma2.it (A.C.); defelici@uniroma2.it (M.D.F.); or gina.la.sala@uniroma2.it (G.L.S.); 4Department of Systems Medicine, University of Rome Tor Vergata, 00133 Rome, Italy; 5Unit of Emergency Room, Emergency Medicine and Internal Medicine, Isola Tiberina-Gemelli Isola Hospital, 00186 Rome, Italy; luigi.carbone@fbf-isola.it; 6CNR Institute of Biochemistry and Cell Biology, 00015 Rome, Italy

**Keywords:** diabetic retinopathy, retinal neurodegeneration, Müller cells, glucose fluctuations, diabetes, progenitor cells, dedifferentiation

## Abstract

Retinal neurodegeneration (RN), an early marker of diabetic retinopathy (DR), is closely associated with Müller glia cells (MGs) in diabetic subjects. MGs play a pivotal role in maintaining retinal homeostasis, integrity, and metabolic support and respond to diabetic stress. In lower vertebrates, MGs have a strong regenerative response and can completely repair the retina after injuries. However, this ability diminishes as organisms become more complex. The aim of this study was to investigate the gliotic response and reprogramming potential of the human Müller cell line MIO-M1 cultured in normoglycemic (5 mM glucose, NG) and hyperglycemic (25 mM glucose, HG) conditions and then exposed to sustained high-glucose and glucose fluctuation (GF) treatments to mimic the human diabetic conditions. The results showed that NG MIO-M1 cells exhibited a dynamic activation to sustained high-glucose and GF treatments by increasing GFAP and Vimentin expression together, indicative of gliotic response. Increased expression of SHH and SOX2 were also observed, foreshadowing reprogramming potential. Conversely, HG MIO-M1 cells showed increased levels of the indexes reported above and adaptation/desensitization to sustained high-glucose and GF treatments. These findings indicate that MIO-M1 cells exhibit a differential response under various glucose treatments, which is dependent on the metabolic environment. The in vitro model used in this study, based on a well-established cell line, enables the exploration of how these responses occur in a controlled, reproducible system and the identification of strategies to promote neurogenesis over neurodegeneration. These findings contribute to the understanding of MGs responses under diabetic conditions, which may have implications for future therapeutic approaches to diabetes-associated retinal neurodegeneration.

## 1. Introduction

Diabetic retinopathy (DR) is the most common and sever microvascular complication of diabetes mellitus (DM), accounting for the majority of vision damage to the retina and blindness in adults [1]. In recent years, the concept of DR as a microvascular disease has evolved, as it is now recognized as a more complex diabetic complication in which retinal neurodegeneration (RN) plays a significant role [2,3]. In this contest, RN has recently been considered as an early marker of DR, since it seems to precede vascular damage [4].

The neurovascular unit (NVU) in the retina refers to the functional coupling and interdependency of neurons (e.g., ganglion cells, amacrine cells, horizontal and bipolar cells), glia (e.g., Müller cells and astrocytes), immune cells (e.g., microglia and perivascular macrophages), and highly specialized vascular cells (e.g., endothelial cells and pericytes) [5]. The impairment of the NVU represents a primary event in the pathogenesis of DR and is characterized by RN and early microvascular alterations. The hallmarks of diabetes-induced RN, which include neural-cell apoptosis, diminished retinal neuronal function, and reactive gliosis, have been observed to occur prior to the onset of overt microangiopathy in experimental models of DR, diabetic patients, and post-mortem human retinas [4,6,7,8].

Müller Glia cells (MGs) are the main retinal glial cells, extending throughout the entire thickness from the inner to the outer limiting membranes, providing an anatomical link between the retinal neurons and the retinal blood vessels and thus responsible for the homeostatic and metabolic support of retinal nerve cells [9]. MGs also play a pivotal role in maintaining neuronal health through the recycling of neurotransmitters and the prevention of neurotoxicity resulting from glutamate excess. Additionally, they regulate ionic balance by buffering K^+^ ions and supply lactate as an energy source for photoreceptors [10]. In response to diabetic stress, MGs undergo morphological changes and exhibit an increased expression of the intermediate filament proteins, such as glial fibrillary acidic protein (GFAP), a key marker of reactive gliosis [10]. Reactive gliosis plays a dual role in the context of retinal damage, with the potential to both protect against damage and contribute to its progression in a diabetic milieu. The formation of a glial scar, while stabilizing the tissue structure, may also impair neuronal function through the release of proinflammatory mediators [11,12]. The ambivalent role of MGs gliosis in DR underscores the complexity of their involvement, thereby emphasizing the necessity for a comprehensive understanding of the regulatory mechanisms involved. The impact of fluctuating glycemic conditions on MGs responses introduces an additional layer of complexity. Fluctuations in blood glucose levels have been demonstrated to trigger diverse signaling pathways within MGs, resulting in an increase of the inner nuclear layer (INL) in subjects with diabetes [13,14], potentially leading to differential activation states or pathological outcomes [10,15,16]. As a matter of fact, in recent years significant advancements have been made in understanding the pathophysiological roles of MGs in DR [17].

Studies in lower vertebrates, including fish and amphibians, have demonstrated that, following reactive gliosis, the sonic hedgehog (SHH) signaling pathway is activated, thereby promoting the reprogramming of MGs into various retinal cell types and contributing to retinal regeneration [18,19]. The capacity for retinal regeneration is notably limited in mammals, including humans. In response to injury, MGs typically undergo a process of reactive gliosis, which often results in scarring rather than regeneration [11,20]. To achieve significant regenerative processes, additional manipulations of the SHH pathway may be required [21,22,23]. Recent studies have identified the potential genes involved in reprogramming MGs into retinal neurons, such as Ascl1, Sox2, and Lin28, which can be modulated to stimulate functional neuronal regeneration [24,25,26]. Despite the advancement of knowledge regarding the neurogenic process, however, the existing literature lacks insights into the influence of glucose metabolism on these processes. In particular, as far as we know, there is no evidence in the literature indicating the effect of glucose metabolism on retinal neurogenesis.

Our previous in vitro studies, involving rat retinal Müller cell line rMC-1, have demonstrated that these cells are activated in response to both high glucose levels and glucose fluctuations (GFs), and that this activation is associated with cellular increased levels of GFAP and aquaporin-4 (AQPs) [27]. These findings contributed to elucidating the dynamic responses of Müller cells under varied glycaemic conditions, suggesting the existence of a complex regulatory mechanism sensitive to metabolic fluctuations. However, to better understand the mechanisms underlying DR in humans, it is essential to investigate whether or not similar responses occur in human Müller cell lines, as cellular responses may differ between animal models and humans, particularly due to differences in complexity and disease progression.

The present study aims to fill this gap by investigating the gliotic response of human Müller cell line MIO-M1, a well-established model in human retinal research, exposed to sustained high-glucose and glucose fluctuation treatments that mimic the different glycemic conditions that can occur in diabetic patients. Furthermore, the study aims to investigate the reprogramming potential of MIO-M1 cells under different glucose stress conditions, focusing on their potential for dedifferentiation and neurogenesis. These processes are of significant interest in understanding how glial cells respond to pathological conditions such as DR. This approach allowed one to explore not only the gliotic response but also the potential for reprogramming Müller cells into a more progenitor-like state, a key aspect for the development of future regenerative therapies. The results obtained might contribute to elucidating the dynamic responses of MGs under different glycemic conditions, suggesting the existence of a complex regulatory mechanism sensitive to glucose metabolic stress.

## 2. Results

### 2.1. HG Condition and Treatments with Sustained High-Glucose and GFs Induce Reactive Gliosis in MIO-M1 Cells

WB analysis carried out on MIO-M1 cells cultured under HG condition showed a significant increase in GFAP levels compared to MIO-M1 cells cultured under NG condition (fold change of about 2 in HG vs. NG), indicating an enhanced gliotic response under hyperglycemic stress conditions (Figure 1A,B).

Such significant increases of GFAP levels was also observed when NG MIO-M1 cells were exposed to sustained high-glucose (II) and GF (III) treatments compared to their basal glucose (I) (Figure 2A,B and Appendix A). Although a clear trend towards increased GFAP expression was also observed across GF treatments IV and V, these increases were not statistically significant (Figure 2A,B).

In contrast, MIO-M1 cells cultured under HG condition showed a markedly different response, with no significant changes in GFAP levels when these cells were exposed to sustained high-glucose (II) and GF (III–V) treatments compared to their basal glucose treatment (I) (Figure 2A,B).

The impact of the experimental culture conditions described above on GFAP expression in MIO-MI cells was also investigated by IF, and cellular bipolar or radial morphologies, typical of quiescent and activated Müller cells, respectively [28,29], were also recorded. Under NG condition, GFAP expression was observed in a limited number of MIO-MI cells, approximately 15%, which were GFAP positive, consistent with literature data [29,30]. Typical of low active Müller cell status, GFAP staining in NG MIO-M1 cells was faint, while bipolar or radial morphologies were equally distributed (Figure 3A,B). In contrast, MIO-M1 cells maintained in the HG condition showed intense GFAP staining, with almost 80% displaying radial morphology (Figure 3A,B), indicating that these cells are in a more activated state.

Similarly to the WB results, in NG MIO-M1cells, subjected to all different glucose treatments (II–V), a significant increase in the GFAP intensity and radial morphology were observed, compared to the basal glucose treatment (I) (NG: 47.85% condition I vs. 69,63% condition II; 77.80% condition III; 74.21% condition IV; 81.42% condition V) (Figure 4A,B). In contrast, in HG MIO-M1 cells subjected to any experimental glucose treatments, no significant change in GFAP positivity or intensity and morphology were detected compared to the basal glucose treatment (Figure 4A,C).

IF for vimentin filaments, another well-established marker of Müller cell stress response, on MIO-M1 cells subjected to all different glucose treatments gave similar results to those described for GFAP and cell morphology changes. Briefly, NG MIO-M1 cells in the basal glucose treatment (I) exhibited intact cytoskeletal integrity, with well-organized and linear vimentin filaments. Following the exposure of NG MIO-M1 cells to sustained high-glucose (II) and GF (III–V) treatments, a significant increase in vimentin expression was observed, compared to the basal glucose treatment (I). This increase was associated with a more disorganized appearance of vimentin filaments, reflecting increased cellular stress and cytoskeletal reorganization (Figure 5). Such a result was also observed in HG MIO-M1 cells maintained in the basal glucose treatment (I) and when these cells were exposed to all different glucose treatments (II–V) (Figure 5).

### 2.2. HG Condition and Treatments with Sustained High-Glucose and GFs Induce Dedifferentiation in MIO-M1 Cells

In order to investigate the activation of the neurogenic potential of MIO-M1 cells under the various glucose treatments investigated here, the expression and cellular localization of sonic hedgehog (SHH), a key signaling protein capable of regulating the differentiation process during retinal development, was assessed [19,31].

WB analysis of MIO-M1 cells maintained under NG and HG conditions revealed a significant increase in SHH protein levels in the latter. IF observations confirmed such results and provided insights into the differential cytoplasmic localization of the protein between NG and HG MIO-M1 cells. In the majority of NG MIO-M1 cells (about 85%), SHH was widespread throughout the cytoplasm. In contrast, in approximately 70% of HG-MIO-M1 cells, although SHH maintained a widespread distribution, it was also predominantly localized in punctate structures (dot-like formations) within the cytoplasm and near the plasma membrane (Figure 6C,D).

The WB results reported in Figure 7A,B show that SHH expression was significantly increased in NG MIO-M1 cells subjected to GF treatments (III–V), compared to their basal glucose treatment (I).

Although a clear trend towards increased SHH expression was also observed in response to sustained high-glucose treatment (II), this increase was not statically significant (Figure 7A,B).

Conversely, HG MIO-M1 cells subjected to treatments II and V showed no change in SHH expression compared to the basal glucose treatment, while a decrease of the SHH protein level in cells exposed to GF treatments III and IV was observed (Figure 7A,B).

IF basically confirmed these findings at the protein level and revealed a distinct cytoplasm localization of the protein in NG MIO-M1 cells exposed to the different glucose treatments (II–V) compared to the basal glucose treatment (I). In these cells, the increased level of SHH expression observed under sustained high-glucose (II) and GF (III–V) treatments was accompanied by a significant change in the intracellular localization of SHH, showing a more punctate pattern, consistent with that of HG MIO-M1 cells (Figure 7C,D). In contrast, HG MIO-M1 cells exposed to all different glucose treatments showed no changes of intracellular SHH distribution compared to the basal glucose treatment (Figure 7C,D). These results underline the notion that Müller glia undergo adaptation or desensitization when cultured in a chronically hyperglycemic environment and subsequently subjected to further glucose-induced stress.

IF observations and q-PCR analyses for SRY-Box Transcription Factor 2 (SOX2), a transcription factor that plays an important role in reprogramming Zebrafish Müller glia [32,33], showed expression changes of this transcription factor parallel to that of SHH. In short, while approximately 55% of NG MIO-M1 cells were positive for the protein, in line with Lawrence et al. (2007) [30], this increased to 80% in HG MIO-M1 cells (Figure 8A,B). q-PCR analysis confirmed a higher SOX2 transcript in these latter in comparison to the former (Figure 8C).

Additionally, q-PCR analysis revealed a significant increase of SOX2 mRNA levels in NG MIO-M1 cells exposed to sustained high-glucose (II) and GF (III–V) treatments compared to the basal glucose treatment (I) (Figure 8D). In contrast, no significant changes in SOX2 mRNA levels were observed in HG MIO-M1 cells when the cells were exposed to the same glucose treatments (Figure 8D).

## 3. Discussion

Müller glial cells play fundamental roles in retinal tissue functions. Therefore, studies into their biology and functions may contribute to understanding the causes of retinal pathologies and to develop strategies to alleviate their outcomes. The aim of this study was to investigate the impact of high glucose and glucose fluctuations on critical cellular processes, such as gliosis and reprogramming in Müller cells. These processes are of pivotal importance in the onset and progression of retinal neurodegeneration and diabetic retinopathy.

In the present study, we observed a significant increase in GFAP expression in NG MIO-M1 cells exposed to sustained high-glucose and GF treatments, indicating a glucose stress-induced gliotic response. This response was associated with other markers of reactive gliosis, such as the morphological changes associated with the transition from a bipolar to a radial morphology and the overexpression/reorganization of vimentin intermediate filament. These findings indicate that NG MIO-M1 cells are sensitive to glucose changes in their surrounding environment, potentially functioning as a protective or reactive mechanism in response to metabolic stress. The remarkable sensitivity to glucose stress and the prompt reactive gliosis observed in MIO-M1 cells were found to be similar to those observed in Müller cells of diabetic subjects [8]. This finding confirms the suitability of MIO-M1 cells as an in vitro model to study Müller cell response. In contrast, HG MIO-M1 cells maintain a basal high level of both GFAP and Vimentin together to less organized intermediate filaments across the different glucose treatments, suggesting a limited and static response to sustained high-glucose and GF treatments. These findings highlight that MIO-M1 cells in chronic hyperglycemic conditions may adapt or develop enhanced tolerance over time due to constant and prolonged exposure to high glucose levels. The adaptation or desensitization of HG MIO-M1 cells to sustained high-glucose and GFs may indicate a saturation of the gliotic response, which is characteristic of prolonged hyperglycemic states, as observed in uncontrolled diabetes [34]. This adaptation or desensitization may result in a reduction in the cellular capacity to respond to additional metabolic challenges or stress.

To understand the clinical consequences of Müller cell gliosis it is essential to also consider the reprogramming process, which may include dedifferentiation of Müller cells and regeneration of various retinal neurons, contributing to retinal tissues repair under certain conditions. In species such as birds [35], zebrafish [36,37], and rodents [38], Müller glia contribute as the primary source of retinal regeneration. When the retina is damaged, Müller glia can dedifferentiate into Müller glia-derived progenitor cells (MGPCs), acquiring a progenitor-like phenotype and starting to proliferate, thus contributing to retinal repair [39,40]. Notably, Müller glia in lower vertebrates exhibit a remarkable ability to regenerate retinal neurons, contrasting with the limited regenerative capacity seen in mammals, including humans [40], where Müller cells typically respond to injury with reactive gliosis, leading to scarring rather than regeneration [11,20]. The SHH signaling pathway is essential for the proper development of all vertebrate retinas [41,42,43,44,45,46,47], and several studies have demonstrated the involvement of SHH signaling in the proliferation and differentiation of MGPCs, contributing to retinal regeneration in lower vertebrate [18,19,31,32,34]. However, in mammals, the SHH pathway alone may not be sufficient to overcome the intrinsic limitations of Müller cells to regenerate neuronal cells.

In our study, we observed an increase in SHH expression in NG MIO-M1cells exposed to sustained high-glucose and GF treatments. This suggests that these cells are engaging in pathways associated with cellular dedifferentiation and potential reprogramming towards a progenitor state. The increased SHH protein expression in NG cells was accompanied by significant changes in the intracellular localization of SHH, exhibiting a more spotted/punctate pattern in the cytoplasm and in close proximity to the plasma membrane. These findings suggest an enhanced state of cellular activity with increased synthesis and potential accumulation of SHH, aligning with the findings in lower vertebrates and rodents, where increased SHH signaling plays a role in Müller cell reprogramming and retinal regeneration [18,48,49]. Furthermore, significantly higher levels of SHH were observed in HG MIO-M1 cells compared to NG MIO-M1 cells. Conversely, HG MIO-M1 cells exposed to GF treatments showed a decrease in SHH protein expression levels. A trend toward a decrease was also observed in response to sustained high-glucose. This finding is consistent with a previous study in which the authors observed a decrease in SHH signaling in reactive astrocytes of the cerebral cortex after acute, focal injury, particularly in cells proximal to the lesion site [50]. This highlights that the negative regulation of SHH activity in astrocytes is context-dependent and varies with the degree of cellular damage. In our study, the decreased SHH levels observed in HG MIO-M1 cells exposed to GFs indicate that the severity of metabolic stress may influence SHH signaling pathways through a similar mechanism. These observations underscore the notion that glial cells, including Müller cells and astrocytes, exhibit differential SHH responses depending on the extent of glucose stress, highlighting the dynamic regulation of SHH signaling in glial cells. Although the SHH level has decreased, the cells that express it maintain a dot-like distribution of SHH inside the cells. Our findings suggest that, under NG condition, exposure to sustained high-glucose and GFs stimulates SHH expression, potentially promoting the dedifferentiation and reprogramming of Müller cells. However, in HG-adapted cells, additional metabolic stress from GF treatments leads to decreased SHH expression, possibly impairing regenerative capacity and enhancing gliotic responses. These observations indicate that the severity and fluctuation of glucose stress influence SHH signaling pathways, affecting the balance between neuroprotection and gliosis in the retina.

A previous study demonstrated that high-glucose conditions have been associated with increased SOX2 levels in human Müller glial cells, which may support cell survival and regeneration under stress conditions [51]. In our experiments, SOX2 was significantly upregulated in HG MIO-M1 cells, compared to NG MIO-M1 cells. Furthermore, NG MIO-M1 cells have the capacity to modulate this gene when exposed to sustained high-glucose and GF treatments, exhibiting a consistent upregulation of SOX2. These findings are consistent with the observed upregulation of SHH in NG MIO-M1 cells exposed to different glucose treatments. Unlike NG MIO-M1cells, HG MIO-M1 cells exposed to the same glucose treatments do not show variations in SOX2 expression. This different behavior of HG MIO-M1 cells, with regards to SOX2 and SHH expression, may be due to the different roles and regulatory mechanisms in the glial cells of these two genes. SOX2 is a transcription factor crucial for maintaining stemness and promoting cell progenitor proliferation, and its upregulation could be a response to cellular stress in order to maintain or enhance regenerative capacity. In contrast, SHH signaling, which is involved in cell differentiation and tissue patterning, may be more sensitive to metabolic perturbations with a more dynamic regulation, which can lead to its downregulation under glucose fluctuations treatments. However, despite the observed upregulation of SHH and SOX2 in NG MIO-M1 cells, the expression of rhodopsin, a marker of mature neuronal cells, was not detected. This indicates that, although the cells possess the potential to differentiate into neuronal cells, the conditions employed in our protocol were not sufficient to fully induce this differentiation. The lack of mature neuronal marker expression is likely due to the timing and duration of the experimental conditions. It may therefore be worthwhile to optimize the protocol in order to more accurately reflect the diabetic conditions in humans, with a view to promoting full neuronal differentiation. Given the role of SHH in Müller cells reprogramming and the observed upregulation under sustained high-glucose and GF treatments, future experiments could investigate the effect of exogenous SHH treatment on human Müller cells. This approach would provide insights into the therapeutic potential of modulating SHH signaling to mitigate the adverse effects of reactive gliosis and enhance regenerative processes in the diabetic retina.

Although the current study provides important insights into the gliotic and reprogramming potential of Müller cells under different glucose treatments, some limitations should be considered. The study employed the MIO-M1 cell line, which, although well established in human retinal research, may not fully capture the physiological complexity of Müller cells found in human retina. Future research could benefit from using primary Müller cells to better reflect the in vivo context of human diabetic retinopathy. In addition, the effects of other stressors, such as oxidative stress or hypoxia, which also play a role in diabetic retinal damage, were not included in our model. Inclusion of these factors could provide a more comprehensive understanding of the cellular mechanisms involved. Despite these limitations, the current study contributes important insights into the gliotic and reprogramming potential of Müller cells under varying glucose conditions.

## 4. Materials and Methods

### 4.1. MIO-M1 Cell Cultures Treatment

The Müller cell line (MIO-M1; YB-H3309, Ybio, Shanghai, China) was maintained under controlled conditions at 37 °C and cultured in Dulbecco’s Modified Eagle Medium (DMEM) (11966-025, Gibco, Grand Isle, VT, USA). The culture medium was supplemented with 10% fetal bovine serum (FBS; Gibco), 5 mg/mL streptomycin, 5 U/mL penicillin (Gibco), and 5 mM (1 g/L) glucose or 25mM (4.5 g/L) glucose. The experimental conditions, referred to as normoglycemic (NG) at 5 mM glucose and hyperglycemic (HG) at 25 mM glucose, represent in humans the normal physiological levels of glucose (100 mg/dL) and the hyperglycemic conditions, respectively [52].

In order to evaluate the influence of sustained high-glucose and glucose fluctuations (GFs) on cellular function, MIO-M1 cultured in NG and HG conditions were plated at a density of 10,000 cells/cm^2^ and exposed to different glucose treatments over 96 h, with media changes every 24 h. Treatments were as follow: Treatment I = constant basal glucose medium (5 mM for NG cells and 25 mM for HG cells); Treatment II = sustained high-glucose medium (25 mM for NG cells and 45 mM for HG cells); Treatment III = alternating basal (5 mM for NG cells and 25 mM for HG cells) and high-glucose medium (25 mM for NG cells and 45 mM for HG cells) every 24 h; Treatment IV = basal glucose medium (5 mM for NG cells and 25 mM for HG cells) for 72 h followed by high-glucose medium (25 mM for NG cells and 45 mM for HG cells) for the last 24 h; Treatment V = alternating low- (3 mM for NG cells and 5 mM for HG cells) and high-glucose (25 mM for NG cells and 45 mM for HG cells) medium every 24 h. A detailed scheme of the cell culture conditions and treatments is provided in Appendix A.

### 4.2. Western Blot (WB)

MIO-M1 cell lysates were prepared with RIPA buffer (25 mM Tris–HCl pH 7.5, 150 mM NaCl, 1% NP-40, 0.1% sodium dodecyl sulphate (SDS), 1% sodium deoxycholate, 10 mM sodium fluoride (NaF), 1 M phenylmethylsulphonyl fluoride (PMSF), 1 M sodium vanadate (NaVO_3_), containing EDTA-free protease inhibitor cocktail (Sigma-Aldrich, St. Louis, MO, USA) and PhosSTOP™ (Roche, Penzberg, Germany). The samples were incubated for 30 min at 4 °C, centrifuged at 13,000 rpm for 10 min at 4 °C, and then sonicated once (5 s, 10% power) [27]. The protein concentration was determined using the Bradford assay.

Protein samples were separated by electrophoresis on 12% (*v*/*v*) sodium dodecyl sulfate polyacrylamide gel electrophoresis (SDS–PAGE) gels and transferred to a polyvinylidene difluoride (PVDF) Transfer Membrane Hybond™ (Amersham Biosciences, Amersham, UK). Membranes were then blocked with 5% non-fat dry milk in phosphate buffered saline (PBS) containing 0.05% (*v*/*v*) Tween 20 (PBS-T) for 1 h at room temperature, followed by overnight incubation at 4 °C with the primary antibodies. The following primary antibodies were used: rabbit anti-GFAP (ab7260, Abcam, Cambridge, MA, USA, 1:5000); goat anti-SHH (ab240438, Abcam 1:250); mouse anti-HSP90 (ab79849, Abcam, 1:1000); rabbit anti-α-tubulin (SC-9104, La Santa Cruz Biotechnology, Dallas, TX, USA, 1:500) diluted with 5% BSA in PBS-T. Secondary HPR-conjugated antibodies (Amersham Biosciences, Uppsala, Sweden) diluted 1:5000 in 1% (*w*/*v*) non-fat dry milk in PBS-T were used to incubate the membrane for 1 h at room temperature. Immunoreactive bands were detected by Amersham™ ECL™ Prime (Amersham Biosciences), according to the manufacturer’s protocol. Chemical luminescent signals were detected using ImageQuant LAS 4000 mini (GE Healthcare, Chicago, IL, USA). Densitometric analysis of the bands was performed using ImageJ software v1.53k. For normalization of protein expression, membranes were stripped using the Re-Blot Plus Mild Solution (10×) (Merck KGaA, Darmstadt, Germany), following the manufacturers’ protocol, and re-probed with rabbit anti-β-tubulin, mouse anti-HSP90.

### 4.3. Immunofluorescence (IF)

MIO-M1 cells grown on a 96-well plate were treated as indicated above. After washing with PBS, they were fixed with 4% paraformaldehyde (PFA) in PBS for 15 min and then permeabilized with 0.2% Triton X-100 in PBS for 10 min. Cells were incubated with blocking buffer (3% BSA and 0.05% Tween-20 in PBS) for 1 h at room temperature, then incubated overnight at 4 °C with the following antibodies: rabbit anti-GFAP antibody (ab7260, Abcam, 1:500); goat anti-SHH antibody (ab240438, Abcam; 1:50); mouse anti-SOX2 antibody (sc-365823, La Santa Cruz Biotechnology, 1:500); rabbit anti-Vimentin antibody (ab45939, Abcam; 1:200). Subsequently, cells were incubated with the following secondary antibodies: anti-rabbit conjugated with Alexa-Fluor-488 or Alexa-Fluor-568, anti-mouse conjugated with Alexa-Fluor-568 and with an anti-goat conjugated with Alexa-Fluor-568 (Invitrogen, Carlsbad, CA, USA, 1:500) diluted in PBS-T for 1 h at room temperature. Following the staining of the nuclei with Hoechst 33242 dye (0.5 µg/mL, Invitrogen, Carlsbad, CA, USA) diluted in PBS, the cells were examined using a Leica DM6000 B (Leica Microsystems, Wetzlar, Germany). Images were captured using the LAS AF acquisition software (2.6.0.7266, Leica Microsystems). For the quantification of MIO-M1 cells, 20 fields at 20× magnification were examined for GFAP, SOX2, and SHH in all glucose conditions. Immunoreactive cells were counted in 3 independent experiments using the ImageJ program v1.53k; data were obtained by counting at least 500 cells for each group in at least three independent experiments.

### 4.4. RNA Extraction and qRT-PCR

RNA was extracted from mouse tissues using Trizol Protocol (TRIzol™ Reagent, Invitrogen, ThermoFisher Scientific, Catalog number: 15596026, Waltham, MA, USA), according to the manufacturer’s recommendations. An amount of 1 μg of RNA was reverse transcribed using random primers and the QuantiTect Reverse Transcription Kit (Qiagen, Hilden, Germany), following the manufacturer’s specifications. Gene expression was measured using iTaq Universal SYBR Green Supermix (Biorad Laboratories, Hercules, CA, USA). Real-time PCR was performed in the LightCycler 96 Real-Time PCR System (Roche Diagnostics GmbH, Mannheim, Germany). We used the following primers for the SOX-2 gene: SOX-2-F: GCTACAGCATGATGCAGGACCA; SOX-2-R: TCTGCGAGCTGGTCATGGAGTT; RHODOPSIN-F: AGCTCGTCTTCACCGTCAAGGA; RHODOPSIN-R: CCAGCAGATCAGGAAAGCGATG. The GAPDH gene was used as a housekeeping gene for all experimental samples: GAPDH-F: TCGGAGTCAACGGATTTGGT; GAPDH-R: GAATTTGCCATGGGTGGAAT. Gene expression levels were calculated using the 2^−ΔCt^ method.

### 4.5. Statistical Analysis

All the results are expressed as the mean ± SEM (standard error of the mean) of at least three independent experiments. Statistically significant differences were assessed using Prism 6.05 (GraphPad PRISM Software, Inc., La Jolla, CA, USA) with Student’s *t*-test for statistical comparison between groups. Differences between means were considered statistically significant when *p*-values were at least <0.05.

## 5. Conclusions

The results showed that MIO-M1 Müller cells exhibit distinct responses to different glucose treatments, which are strongly dependent on their metabolic environment. This differential response could have implications for the onset and progression of diabetic retinopathy. The increased levels of activation and dedifferentiation markers observed in NG MIO-M1 cells in response to different glucose treatments suggest a protective response that may occur in early or well-controlled diabetes. In contrast, the lack of response observed in HG MIO-M1 cells exposed to different glucose treatments may reflect an exhausted gliotic and reprogramming capacity, which could contribute to the development and progression of diabetic complications such as retinal neurodegeneration and diabetic retinopathy observed in uncontrolled diabetic patients exposed to prolonged metabolic glucose stress.

Although our results provide valuable insights, further in-depth studies are needed to elucidate the mechanisms involved and to explore the potential therapeutic implications for mitigating retinal neurodegeneration associated with diabetes. Extending these studies to primary Müller cells in future research would provide a more physiologically relevant system, helping to validate and refine our findings. This approach would provide a deeper understanding of the cellular response to sustained high-glucose and glucose fluctuations and could improve the development of targeted therapeutic strategies for retinal pathologies in human patients.

## Figures and Tables

**Figure 1 ijms-25-12877-f001:**
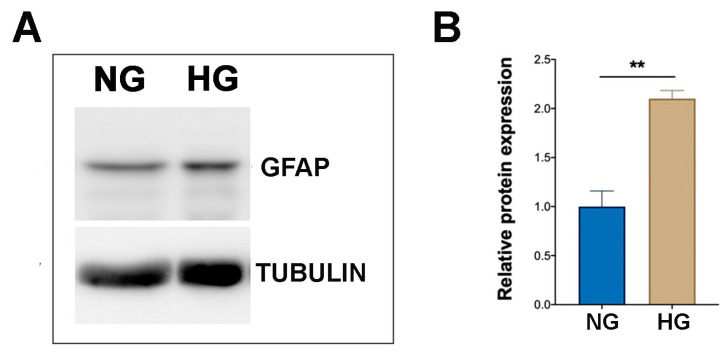
Effect of NG and HG culture conditions on GFAP expression in MIO-M1 cells. (**A**) Representative WB displaying GFAP band obtained from protein extracts of MIO-M1 cells maintained in NG (5 mM glucose, NG cells) and HG (25 mM glucose, HG cells) culture media. α-TUBULIN was used as a loading control. (**B**) Densitometric analysis of GFAP expression, normalized to α-TUBULIN. The ratio of GFAP protein to α-TUBULIN in HG cells was statistically compared to NG cells. Data are presented as mean ± SEM from three independent experiments. Statistical significance (** *p* < 0.01 vs. NG cells) was determined using Student’s *t*-test.

**Figure 2 ijms-25-12877-f002:**
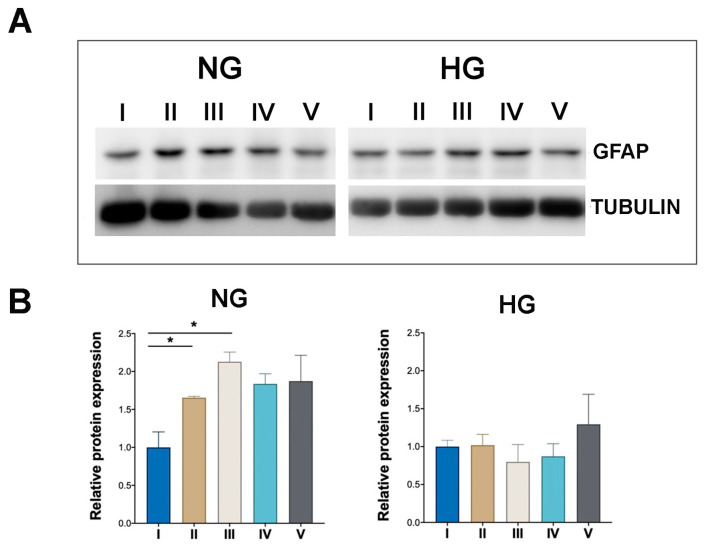
Effect of sustained high-glucose and GF treatments on GFAP expression in NG and HG MIO-M1 cells. NG and HG MIO-M1 cells were exposed to sustained high-glucose and GF treatments as described in M&M and in Appendix A. (**A**) Representative WB analysis showing GFAP protein levels in NG and HG MIO-M1 cells subjected to different glucose treatments (I–V); α-TUBULIN was used as loading control. (**B**) Densitometric analysis of GFAP expression normalized to α-TUBULIN. The ratio of GFAP protein to α-TUBULIN in NG and HG cells subjected to all different glucose treatments was compared to cells cultured under basal glucose treatment (I). Data are presented as mean ± SEM from three independent experiments. Statistical significance (* *p* < 0.05) was determined by Student’s *t*-test.

**Figure 3 ijms-25-12877-f003:**
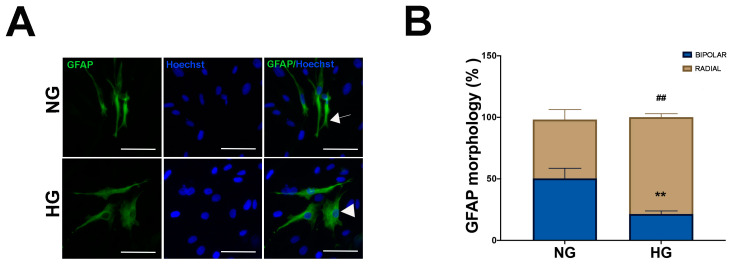
Effect of NG and HG culture conditions on GFAP expression and morphology in MIO-M1 cells. MIO-M1 were maintained in NG (5 mM glucose, NG cells) and HG (25 mM glucose, HG cells) culture media. (**A**) Representative photomicrographs of NG (upper panel) and HG (lower panel) MIO-M1 cells showing IF for GFAP (green). Nuclei were stained with Hoechst (blue). Bipolar and radial cells are indicated by an arrow and an arrowhead, respectively. Scale bar: 75 μm. (**B**) Plots showing the percentage of GFAP-positive cells with bipolar and radial morphology in NG and HG MIO-MI cells. Data were obtained by counting at least 500 cells for each group. Data are presented as mean ± SEM from three independent experiments. Statistical significance (** *p* < 0.01 vs. bipolar cells of NG cells; ## *p*  <  0.01 vs. radial cells of NG cells) was determined using Student’s *t*-test.

**Figure 4 ijms-25-12877-f004:**
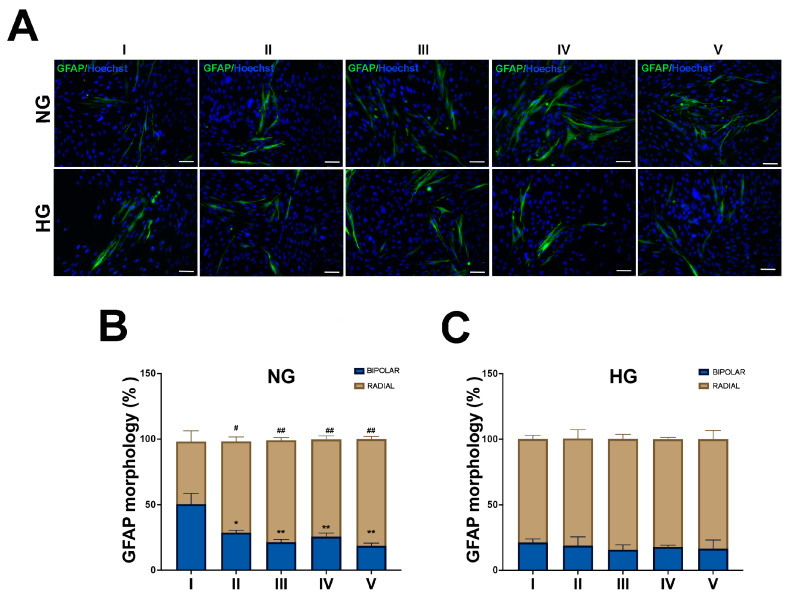
Effect of sustained high-glucose and GFs on GFAP expression and morphology in NG and HG MIO-M1 cells. NG and HG MIO-M1 cells were exposed to sustained high-glucose and GF treatments, as described in M&M and in Appendix A. (**A**) Representative photomicrographs of NG (upper panel) and HG (lower panel) MIO-M1 cells showing IF for GFAP (green). Nuclei were stained with Hoechst (blue). Scale bar: 75 μm. (**B**,**C**) Plots showing the percentage of GFAP-positive cells with bipolar and radial morphology in NG and HG MIO-MI cells exposed to different glucose treatments (I–V). Data were obtained by counting at least 500 cells for each group. Data are presented as mean ± SEM from three independent experiments. Statistical significance (* *p* < 0.05 and ** *p* < 0.01 vs. bipolar cells of treatment I; # *p* < 0.05 and ## *p* < 0.01 vs. radial cells of treatment I) was determined using Student’s *t*-test.

**Figure 5 ijms-25-12877-f005:**
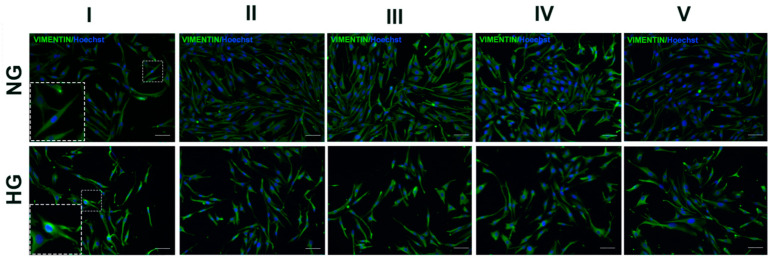
Effect of sustained high-glucose and GFs on Vimentin filaments of NG and HG MIO-M1 cells. NG and HG MIO-M1 cells were exposed to sustained high-glucose and GF treatments (I–V), as described in M&M and in Appendix A. Representative photomicrographs of NG (upper panel) and HG (lower panel) MIO-M1 cells showing IF for Vimentin (green). Nuclei stained with Hoechst (blue). The insets show the organization of Vimentin filaments. Scale bar: 75 μm.

**Figure 6 ijms-25-12877-f006:**
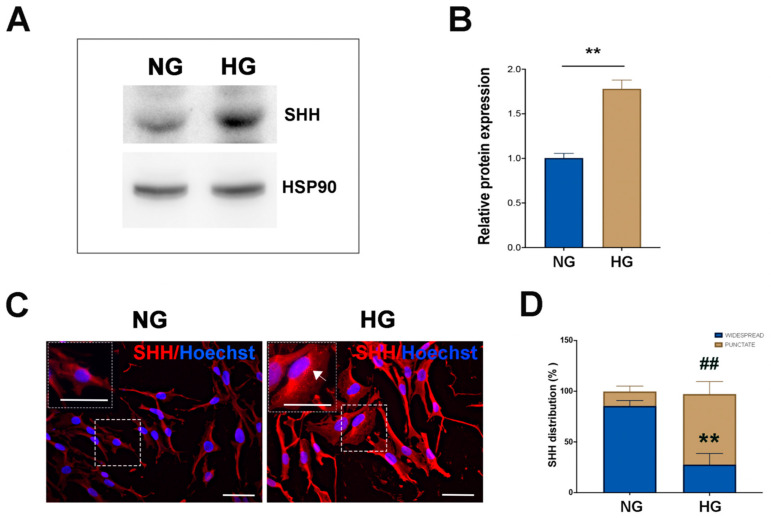
Effect of NG and HG culture conditions on SHH expression in MIO-M1 cells. (**A**) Representative WB displaying basal SHH levels in MIO-M1 cells cultured either in NG (5 mM glucose) or HG (25 mM glucose) media. HSP90 was used as a loading control. (**B**) Densitometric analysis of SHH expression normalized to HSP90. The ratio of SHH protein to HSP90 in HG cells was statistically compared to NG cells. Data are represented as mean ± SEM from three independent experiments. Statistical significance (** *p* < 0.01 vs. NG cells) was determined using Student’s *t*-test. (**C**) Representative micrographs showing SHH (red) in MIO-M1 cultured under NG and HG conditions. Hoechst was used for nuclei staining. The inset highlight SHH inside the cells, with spots marked by an arrow. Scale bar: 20 μm. (**D**) Quantitative analysis of SHH distribution inside the cells was performed using the ImageJ program v1.53k. Data were obtained by counting at least 500 cells for each group. Data are presented as mean ± SEM from three independent experiments. Statistical significance (** *p* < 0.01 vs. widespread NG cells; ## *p* < 0.01 vs. punctate NG cells) was determined using Student’s *t*-test.

**Figure 7 ijms-25-12877-f007:**
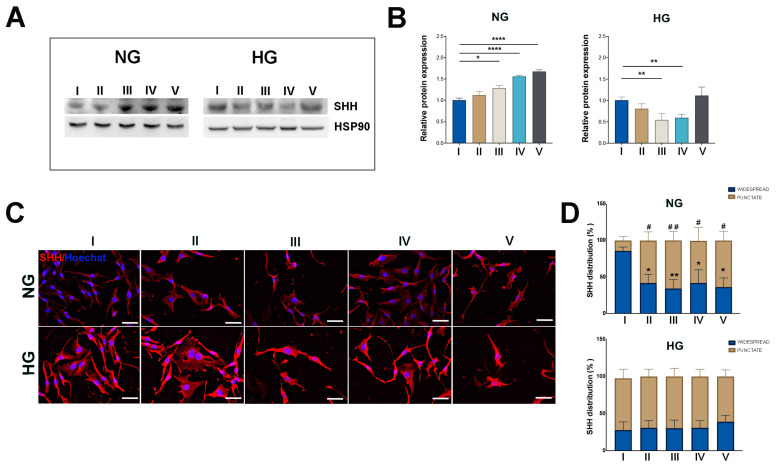
Effect of sustained high-glucose and GFs on SHH expression in NG and HG MIO-M1 cells. NG and HG MIO-M1 cells were exposed to sustained high-glucose and GF treatments as described in M&M and in Appendix A. (**A**) Representative Western blot analysis showing SHH protein levels in MIO-M1 cells cultured in NG and HG conditions subjected to different glucose treatments (I–V). HSP90 was used as a loading control. (**B**) Densitometric analysis of SHH expression normalized to HSP90 for both NG and HG cells in the different treatments (I–V). The ratio of SHH to HSP90 in NG and HG cells subjected to all different glucose treatments was compared to cells cultured under the basal glucose treatment (I). Data are presented as mean ± SEM from three independent experiments. Statistical significance (* *p* < 0.05; ** *p* < 0.01; **** *p* < 0.0001) was determined by Student’s *t*-test. (**C**) Representative micrographs showing SHH (red) in MIO-M1 cells cultured in NG (upper panel) and HG (lower panel) and exposed to different glucose treatments (I–V). Scale bar: 20 μm. (**D**) Quantitative analysis of SHH distribution within the cells was performed using the ImageJ program v1.53k. Plots represent the percentage of SHH-positive cells exhibiting a widespread and punctate morphology among all SHH-positive cells counted in the NG and HG cells exposed to different glucose treatments (I–V). Data were obtained by counting at least 500 cells for each group. Data are presented as mean ± SEM from three independent experiments. Statistical significance (* *p* < 0.05 and ** *p* < 0.01 vs. widespread NG cells in the basal glucose treatment (I); # *p* < 0.05; ## *p* < 0.01 vs. punctate NG cells in the basal glucose treatment (I)) was determined using Student’s *t*-test.

**Figure 8 ijms-25-12877-f008:**
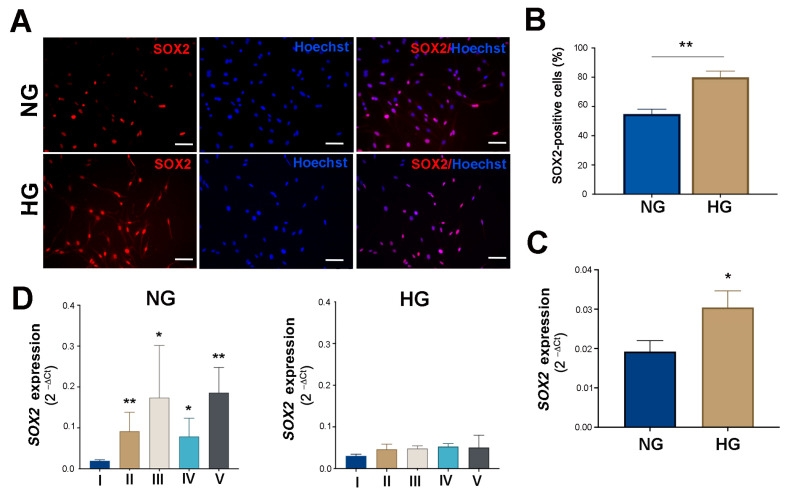
Effect of NG and HG culture conditions and of sustained high-glucose and GF treatments in NG and HG MIO-M1 cells on SOX-2 expression. MIO-M1 Müller cells cultured in NG and HG conditions and under different glucose treatments for 96 h (I–V), as described in Appendix A. (**A**) Representative micrographs of MIO-M1 cultured in NG (upper panel) and HG (lower panel) conditions showing the immunostaining for SOX-2 (red). Nuclei were stained with Hoechst (blue). Scale bar: 20 μm. (**B**) Quantitative analysis of SOX-2-positive cells was performed using the ImageJ program v 1.53k. Plots represent the percentage of SOX-2-positive cells relative to all Hoechst-positive cells counted in NG and HG culture conditions. Data were obtained by counting at least 500 cells for each group. Data are presented as mean ± SEM from three independent experiments. (**C**) qRT-PCR was performed on MIO-M1 cells cultured in NG and HG conditions and SOX-2 transcript analysis was obtained with 2^−ΔCt^ methods. (**D**) qRT-PCR was performed on MIO-M1 cells cultured in NG and HG media under different glucose treatments (I–V), and SOX-2 transcript analysis was obtained with 2^−ΔCt^ methods. Data are presented as mean ± SEM from three independent experiments. Statistical significance (* *p* < 0.05, ** *p* < 0.01) was determined by Student’s *t*-test.

## Data Availability

The datasets and materials used and/or analyzed during the current study are available from the corresponding author upon reasonable request.

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
