# Peer review of "Gliotic Response and Reprogramming Potential of Human Müller Cell Line MIO-M1 Exposed to High Glucose and Glucose Fluctuations†"

_ijms, 2024, doi:10.3390/ijms252312877_

Round 1

Reviewer 1 Report

Comments and Suggestions for Authors

The manuscript is interesting. But, there are several limitations to conclude the study.

Using one human Müller cell line MIO-M1 is not ideal. Several cell lines should be used to generalize the current outcome. Or primary cells should be also considered.

Direct single glucose treatment has been used in many years to study diabetic retinopathy in vitro to mimic hyperglycemia. But, it may not be the best approach. Recently, many metabolites systemically affect the eye/retina. And some groups are using oxidative stress inducers or hypoxic stress inducers or some combination conditions, as single glucose treatment may not be enough relevant to mimic the in vivo system. This should be cautious. Shortly, listing the data (phenotypes) after glucose treatment may not be effective to apply to the in vivo system's conclusion in the future.

What pathways are involved should be clearly detected with genetic modulation in vitro.

Author Response

We would like to thank the reviewer for his constructive feedback. We fully acknowledge the points raised and have carefully considered each of them. Below we provide our responses to the reviewer's comments, addressing each point in detail to clarify our approach and the changes made to the manuscript.

 Using one human Müller cell line MIO-M1 is not ideal. Several cell lines should be used to generalize the current outcome. Or primary cells should be also considered

We appreciate the reviewer comment and agree with the limitation of using a single cell line, MIO-M1, in our study. While it is true that multiple cell lines or primary cells can perhaps provide a more comprehensive understanding, we selected MIO-M1 cells for several reasons. Such cell line has been extensively characterized and used in retinal research, particularly in studies related to diabetic retinopathy and in studies where Müller cells respond to glycemic stress conditions (Lawrence et al., 2007; Zong et al., 2010; Vellanki et al., 2016, Limb et al., 2002, Guo et al., 2021). The choice of the MIO-M1 line proved crucial for our study, as it retains many of the physiological properties of primary cells, allowing us to study responses to variable glycemic conditions and hyperglycemia in a controlled in vitro system. While primary cells would indeed be ideal, the difficulty in obtaining them from human donors and the variability in cellular responses could limit data consistency and reliability. Given these challenges, we considered MIO-M1 cells to be a suitable compromise. This cell line offers stability and accessibility, allowing experiments to be repeated without the problems associated with primary samples. We agree that future studies using multiple cell lines or primary cells would increase the validity and extend the application of our findings. We plan to include such approaches in future research to broaden the scope of our conclusions.

In response to your comment, we included in the discussion the limitations of using an in vitro model of Muller cell line and the potential advantages of using primary Müller cells or other human cell lines in future studies. These additions are highlighted in yellow in the Discussion section of the manuscript.

Direct single glucose treatment has been used in many years to study diabetic retinopathy in vitro to mimic hyperglycemia. But, it may not be the best approach. Recently, many metabolites systemically affect the eye/retina. And some groups are using oxidative stress inducers or hypoxic stress inducers or some combination conditions, as single glucose treatment may not be enough relevant to mimic the in vivo system.

We appreciate the comment regarding the use of single glucose treatment and agree with the reviewer that the inclusion of additional metabolic stressors, such as oxidative stress inducers or hypoxic conditions, may increase the physiological relevance of future studies. We are currently considering experiments in which the same Müller cells (MIO-M1) are simultaneously exposed to both glucose and free fatty acids, such as palmitic acid, which are key metabolites involved in the oxidative and inflammatory stress associated with diabetes.  We agree with the reviewer that studying their combined effect will provide deeper insights into the pathophysiological mechanisms underlying diabetic retinopathy and potentially identify novel therapeutic targets.

To address the reviewer comment, we would like to clarify that the primary focus of our study was to investigate Müller cell responses to glucose-induced stress, specifically glucose fluctuations (GF), which better reflects the fluctuating glucose levels typical in diabetic patients. Previous studies have demonstrated that GF, compared to sustained hyperglycemia, intensifies oxidative stress, inflammation, and cellular dysfunction, making it a crucial factor in the progression of diabetic retinopathy (Hsu CR et al., 2015; Lu et al., 2019). Thus, instead of only focusing on sustained glucose treatment alone, we used different glucose protocols, including sustained glucose and glucose fluctuating conditions (treatments III-V), to better mimic the metabolic stress of the diabetic retina, as described in the Methods section and Figure S1 of our manuscript. This approach allows us to isolate and better understand the specific effects of glucose stress on Müller cell behaviour in vitro, while taking into account the complexity of diabetic conditions. Future studies may benefit from incorporating the combined glucose-fatty acid approach, which will provide a more comprehensive understanding of the metabolic perturbations involved in the disease. We discussed this comment in the Discussion section. The added paragraphs are highlighted in yellow in the manuscript.

Shortly, listing the data (phenotypes) after glucose treatment may not be effective to apply to the in vivo system's conclusion in the future. What pathways are involved should be clearly detected with genetic modulation in vitro.

We appreciate the comment regarding the application of in vitro glucose treatment phenotypes to in vivo systems. While we obviously agree that in vitro models, such as the one used in our study, cannot fully replicate the complexity of an in vivo system, they do provide essential insights into the molecular and cellular mechanisms that occur under specific experimental conditions. In this case, by using different glucose treatments that simulate both sustained high glucose and glucose fluctuations, we aim to isolate the direct effects of glucose-induced stress on Müller cell behavior that are central to understanding diabetic retinopathy. The dynamic glucose treatments used in our study were designed to closely mimic the metabolic stress conditions observed in diabetic patients, where glucose fluctuations are often as damaging as sustained hyperglycemia. These phenotypic changes and molecular responses in a controlled in vitro system are critical for identifying potential pathways and targets for therapeutic intervention. It is evident we that in vivo validation is necessary, the findings from this in vitro model can provide a useful basis for identifying potential pathways and mechanisms involved in glial activation and neurodegeneration that could be further tested in animal models.

While our current study focused on the expression of key markers such as GFAP, SHH, and SOX2, we recognize that genetic modulation could further refine our understanding of the underlying mechanisms. As part of our study, we also performed experiments to assess the involvement of the ERK signaling pathway, which has been previously implicated in glial activation (Picconi et al., 2019). However, our results did not show significant differences in ERK activation under the glucose fluctuation treatments, suggesting that ERK may not be the key mediator in this model. We plan to explore additional pathways in future studies, such as PI3K/AKT or NF-kB, to better understand the molecular mechanisms that govern diabetic retinopathy.

Reviewer 2 Report

Comments and Suggestions for Authors

Current report investigated the gliotic response and reprogramming potential of the human Müller cell line MIO-M1 cultured in various glucose condition. Results and findings seem interesting and useful. Please conduct the concerns below.

1.      Difference of current findings with the previous report [27] was not introduced in clear.

2.      Sample size in each group seems too small (N=3 only). Please describe it in detail.

3.      In Western blot, reliable reference(s) may support the data.

4.      The decreased SHH levels observed in HG MIO-M1 cells exposed to GFs were same as changes in diabetic retinopathy? Please add reference(s) to support it.

5.      The role of SHH in Müller cells reprogramming needs to discuss in clear.

6.      Conclusion section is required to add in revised version.

7.      Limitation(s) of current report seems helpful.

Author Response

Difference of current findings with the previous report [27] was not introduced in clear.

We appreciate the reviewer’s comment and acknowledge the need for a clear distinction between our current findings and the previous report. To address this, we have expanded and clarified the differences in the Introduction of the manuscript. This extended part has been added in yellow in the revised manuscript for easy identification.

Sample size in each group seems too small (N=3 only). Please describe it in detail.

We thank the reviewer for pointing out the sample size concern. To clarify, the sample size n=3 refers to three independent experiments with each experiment consisting of three replicates for each treatment.

The data from each independent experiment were averaged to calculate the mean ± SEM. This information has been specified in the Figure Legend. This corrected part has been added in yellow in the revised manuscript for easy identification.

 In Western blot, reliable reference(s) may support the data

We appreciate your suggestion regarding the Western blot references. In response, we have added the reference of our previous work, which supports the use of the Western blot data in this context. Specifically, the reference cited is from our earlier publication (Picconi et al., 2019), where similar Western blot protocols and methods were applied to assess the expression of GFAP and other markers in retinal cells under glucose treatment conditions."

 The decreased SHH levels observed in HG MIO-M1 cells exposed to GFs were same as changes in diabetic retinopathy? Please add reference(s) to support it. Include additional references or discuss the literature supporting your observation. This will strengthen your interpretation of the decreased SHH levels under these conditions and how it relates to diabetic retinopathy.

The role of SHH in Müller cells reprogramming needs to discuss in clear. more detailed explanation of SHH's involvement in Müller cell reprogramming. Clarify how SHH might influence the gliotic and neurogenic responses under different glucose conditions, possibly referencing previous studies on SHH signaling and Müller cell reprogramming.

Thank you for your constructive feedback. We have carefully revised the manuscript to address your comments. We have expanded the Discussion of the involvement of SHH in Müller cell reprogramming and we added references to key studies showing its impact on gliotic and neurogenic responses under different glucose conditions. You can find these additions highlighted in yellow in the discussion section.

Conclusion section is required to add in revised version. Add conclusions

Thank you for your suggestion regarding the addition of a conclusion section. We have now included a conclusion section in the manuscript highlighted in yellow.

 Limitation(s) of current report seems helpful.

Thank you for your comment, we appreciate your valuable input. In response to your comment, we have added a discussion of the limitations of our study in the discussion section, where we address the use of a single cell line, the potential limitations of not using primary Müller cells, and the exclusion of other stressors, such as oxidative stress and hypoxia. These limitations are now clearly outlined, and we have highlighted the need for future studies to incorporate these aspects in order to provide a more comprehensive understanding of the mechanisms involved. You can find these additions highlighted in yellow in the Discussion section.

Round 2

Reviewer 1 Report

Comments and Suggestions for Authors

The raised comments and requests are at least addressed with listing limitations or mentioning future studies.

Short conclusions are made based on the current data with limited information.

The study claimed that this conclusion would provide new insights into therapeutic strategies for mitigating retinal neurodegeneration associated with diabetes. However, it might be difficult to claim this based on the current dataset. Therefore, at least this part should be written tone-down.

As authors mentioned that all raised requests will be conducted or considered for the future study, it will be interesting to see the future work.

Author Response

Comment

"The study claimed that this conclusion would provide new insights into therapeutic strategies for mitigating retinal neurodegeneration associated with diabetes. However, it might be difficult to claim this based on the current dataset. Therefore, at least this part should be written tone-down."

Response

Thank you for your helpful feedback. We agree that the language used may have overstated the implications of our findings. In response, we have revised the manuscript to tone down such statements and make sure that our conclusions accurately reflect the impact of our data. You’ll find the revised statements in the Abstract and Discussion sections, highlighted in green for easy identification.

Comment

"As authors mentioned that all raised requests will be conducted or considered for the future study, it will be interesting to see the future work."

 Response

Thank you for your feedback. We acknowledge the importance of the requests you have raised and will address them in our future studies. We hope to submit a manuscript that includes the findings from these new studies.

Reviewer 2 Report

Comments and Suggestions for Authors

It has been revised in a good way.

Author Response

Comment

It has been revised in a good way.

Response

Thank you for your positive feedback.